# The Diversity of Encephalitogenic CD4+ T Cells in Multiple Sclerosis and Its Animal Models

**DOI:** 10.3390/jcm8010120

**Published:** 2019-01-19

**Authors:** Benjamin M. Segal

**Affiliations:** 1Holtom-Garrett Multiple Sclerosis Center, Department of Neurology, University of Michigan, Ann Arbor, MI 48109, USA; bmsegal@umich.edu; Tel.: +1-734-615-5635; 2Neurology Service, Ann Arbor VA Healthcare System, Ann Arbor, MI 48105, USA

**Keywords:** Multiple Sclerosis, experimental autoimmune encephalomyelitis, T-helper cells, cytokines

## Abstract

Autoreactive CD4+ T cells, which target antigens in central nervous system (CNS) myelin, are widely believed to play a critical role in the pathogenesis of multiple sclerosis (MS) in concert with other immune effectors. This theory is supported by data from animal model experiments, genome-wide association studies, and immune profiles of individuals with MS. Furthermore, disease modifying agents that target lymphocytes significantly reduce the rate of MS clinical exacerbations. However, the properties of myelin-reactive CD4+ T cells that are critical for their pathogenic activities are not understood completely. This article reviews the literature on encephalitogenic CD4+ T cells, with an emphasis on T-helper (Th) lineage and cytokine production. An increased understanding of the spectrum of encephalitogenic T cells and how they differ from protective subsets is necessary for the development of the next generation of more effective and safer immunomodulatory therapies customized for individuals with MS and related disorders.

## 1. Introduction

Multiple Sclerosis (MS), an inflammatory demyelinating disease of the central nervous system (CNS), is the most common cause of non-traumatic neurological disability among young adults in the Western Hemisphere [1]. Although MS is often portrayed as a disease of North America and Europe, there is increasing evidence that it is more common in other regions of the world (such as Asia, the Middle East, and South America) than previously thought [2,3,4]. MS most commonly presents with a relapsing-remitting course, characterized by self-limited episodes of neurological dysfunction (relapses or exacerbations) that are separated by symptomatically quiescent periods (remissions). To satisfy consensus criteria for a diagnosis of relapsing-remitting MS (RRMS), there must be clinical and/or radiological evidence of two or more lesions in distinct locations (dissemination in space (DIS)) [5]. In addition to DIS, a definitive diagnosis has traditionally required the occurrence of a second clinical exacerbation following the presenting episode or the subsequent emergence of a new lesion visualized by magnetic resonance imaging (MRI) (dissemination in time (DIT)). Recently, the consensus criteria were expanded such that the presence of unique oligoclonal bands in the cerebrospinal fluid (indicative of CNS immunoglobulin production, discussed in detail below) can substitute for DIT [6]. The symptoms associated with MS exacerbations are diverse and commonly include monocular visual loss, limb weakness, different patterns of tingling and/or numbness, incoordination of movement, and impaired gait. Fatigue, double vision, vertigo, and neuropathic pain are also prevalent. This clinical heterogeneity reflects the fact that lesions can arise in any white matter tract throughout the CNS. Lesions can also form in the cerebral cortex and deep grey matter nuclei, because those regions are traversed by myelinated axons. Cortical and deep gray matter volume loss in MS correlates strongly with cognitive impairment [7].

MS relapses are associated with the formation of new inflammatory demyelinating lesions. Acute MRI lesions are identified by the seepage of the magnetic isotope gadolinium into the inflamed tissue, consequent to a focal increase in the blood–brain barrier permeability. Histological analysis of active MS lesions, obtained from biopsy or postmortem specimens, reveals perivascular inflammatory infiltrates, demyelination, and axonal swelling and transection. The infiltrates are composed of lymphocytes and myeloid cells, primarily T cells and monocytes/macrophages. Cerebrospinal fluid (CSF) from the majority of MS patients contains immunoglobulins that are not present in paired serum samples (referred to as oligoclonal bands (OCBs)), indicative of antigen-driven antibody production within the CNS. The presence of unique OCBs in the CSF is an independent predictive factor for the evolution of an initial demyelinating episode (often referred to as a clinically isolated syndrome) into RRMS [8]. The cellular sources of OCBs are presumably meningeal B cells and plasma cells, which have been observed to be clustered with T cells in loose aggregates or lymphoid follicles, in relapsing and progressive MS specimens, respectively [9,10]. By secreting chemokines, such as CXCL12 and CXCL13, and growth factors, such as B cell activating factor (BAFF) and a proliferation-inducing ligand (APRIL), activated microglia and astrocytes could create a CNS microenvironment in people with MS that is conducive to the accumulation and survival of memory B cells and plasma cells [11]. Although the current review article is focused on T cells, B cells play an essential role in MS pathogenesis, as demonstrated by the efficacy of B-cell depleting drugs in suppressing annualized relapse rates [11,12,13,14,15,16]. The mechanism of action of B cells in MS remains to be elucidated. They could act as antigens presenting cells to encephalitogenic T cells, secrete pro-inflammatory and neurotoxic factors, and/or drive the development of meningeal follicles [17].

## 2. The Evidence that MS Is Mediated by Myelin-Reactive CD4+ T Cells

### 2.1. The Autoimmune Hypothesis

The autoimmune theory of MS was initially informed by the advent of the animal model of experimental autoimmune encephalomyelitis (EAE). In the late nineteenth century, a severe, frequently fatal, neurological syndrome occasionally occurred in temporal relation to the administration of a rabies vaccine that was prepared from infected rabbit brains. Analysis of postmortem CNS specimens from afflicted patients revealed histopathological features reminiscent of MS [18]. It was subsequently discovered that the syndrome resulted from contamination of the vaccine with heterologous myelin [19]. The direct vaccination of non-human primates with myelin extracts recapitulated the syndrome [19]. This led to speculation that an autoreactive response against myelin antigens might drive MS pathogenesis. EAE has since been induced in a wide range of mammalian species, most commonly mice, via active immunization with whole myelin proteins or major histocompatibility class II (MHC II)-restricted myelin epitopes in combination with adjuvants. Importantly, EAE can be induced by the adoptive transfer of myelin epitope-primed and -reactivated CD4+ T-cell lines or clones into naïve syngeneic hosts [20]. Thus, the introduction of highly purified myelin-reactive CD4+ T cells is sufficient to trigger an inflammatory demyelinating disease with similar histopathological and clinical features to MS. Although less commonly reported than encephalitogenic CD4+ T cells, myelin-reactive CD8+ T-cell lines have also been shown to be capable of inducing EAE [21]. In some cases, perivascular infiltrates in MS lesions contain CD8+ T cells and CD4+ T cells in close proximity with myeloid cells, suggesting that these subsets act synergistically [22].

Consistent with an autoimmune etiology of MS, several laboratories have discovered that OCBs bind autoantigens as opposed to viral or bacterial antigens [23]. Furthermore, genome-wide association studies (GWAS) have identified over 200 MS susceptibility loci, the majority of which are immune related; many overlap with loci associated with other autoimmune diseases [24]. The strongest association is with genes that encode MHCII molecules, which are necessary for the activation of CD4+ T lymphocytes by antigen-presenting B cells and myeloid cells [25,26]. Single nucleotide polymorphisms (SNPs) in genes encoding the T-cell survival/growth cytokines, interleukin (IL)-2 and IL-7, are also associated with MS risk [27,28]. Collectively, the genetic data strongly suggest that CD4+ T cells play a critical role in MS pathogenesis.

Perhaps the most convincing evidence that myelin-reactive CD4+ T cells are capable of mediating inflammatory demyelination in humans comes from clinical trials of immunomodulatory agents. In a clinical trial published in 2000, RRMS patients were treated with an altered peptide ligand (APL) of myelin basic protein (MBP) with the intention of tolerizing pathogenic MBP-reactive T cells or deflecting their differentiation towards an immunosuppressive, regulatory, or innocuous T-helper 2 (Th2) phenotype [29]. Unexpectedly, the administration of the APL was associated with the expansion of circulating MBP-reactive, IFNγ-producing CD4+ T cells and clinical worsening in a subgroup of patients. Conversely, phase 3, placebo-controlled clinical trials have shown that drugs that impede lymphocyte trafficking to the CNS (i.e., natalizumab and fingolimod), block growth factor signaling into lymphocytes (daclizumab) or deplete lymphocytes from the periphery (alemtuzamab), significantly suppress MS relapse rates and the accumulation of MRI lesions [30,31,32,33,34,35,36,37].

### 2.2. Endogenous Populations of Myelin-Reactive CD4+ T Cells in Individuals with MS

A logical question prompted by the above discussion is whether MS patients harbor an aberrant population of CD4+ T cells capable of reacting against myelin antigens. In fact, myelin-responsive CD4+ T cells have been repeatedly detected among peripheral blood mononuclear cells (PBMCs) and cerebrospinal fluid cells collected from individuals with MS [38]. Surprisingly, circulating myelin-reactive CD4+ T cells have also been isolated from age- and sex-matched healthy controls [39,40]. There is some evidence that potentially encephalitogenic T cells are suppressed in healthy individuals by competent regulatory T cells (Tregs), whereas Tregs are dysfunctional in individuals with MS [41]. Studies that directly compare the frequencies of circulating myelin-reactive T cells in MS patients versus age- and sex-matched healthy controls or controls with non-inflammatory neurological disorders have yielded conflicting results. Earlier studies, which used lymphoproliferation and/or interleukin (IL)-2 production to measure T-cell responses, often found similar frequencies of myelin-reactive PBMCs in subjects with MS versus the control groups [39,40,42]. More recent studies have focused on autoantigen-specific production of effector cytokines, such as IL-17 and IFNγ, arguably a more accurate reflection of the pathogenic response (discussed in Section 4.). Several independent laboratories have reported that untreated RRMS patients have elevated frequencies of PBMC that produce IFNγ or IL-17 in response to ex vivo challenges with human myelin basic protein (MBP), human proteolipid protein (PLP), or their constituent peptides [43,44,45,46]. In one of those studies, IFNγ responses to PLP peptides correlated with the level of clinical disability [45].

## 3. The Characteristics of Encephalitogenic CD4+ T Cells

### 3.1. T-helper (Th) Lineage and Encephalitogenicity

As mentioned above, the adoptive transfer of highly purified myelin-specific CD4+ T cells can trigger EAE in naïve syngeneic mice. However, not all myelin-reactive CD4+ T cells are capable of disease induction. For example, murine myelin-reactive Th2 cells or Th lineage-uncommitted effector T cells do not induce neuroinflammation or clinical deficits following adoptive transfer into immunocompetent wildtype hosts [47,48,49,50]. Th1 cells were originally thought to be the critical autoreactive T-cell subset in CNS autoimmune diseases, because IFNγ is plentiful in active EAE and MS lesions and perivascular infiltrates in both laboratory animals and humans are generally monocyte/macrophage enriched, reminiscent of a Th1-driven inflammatory response [51,52]. Furthermore, myelin-specific T cells isolated from MS patients and immunized mice tended to produce IFNγ but not IL-4 [53]. As mentioned earlier, the expanded populations of MBP-reactive CD4+ T cells in MS patients who experienced severe clinical exacerbations upon experimental treatment with an APL were strongly skewed toward the Th1 lineage [29]. However, the relevance of Th1 cells to autoimmune demyelinating disease was challenged when C57BL/6 mice genetically deficient in IFNγ or in Th1-polarizing molecules (namely the IL-12p35 chain or IL-12 receptor β2 chain) were found to be susceptible to EAE [54,55,56,57,58]. This paradox was apparently resolved when an alternative lineage of myelin-reactive CD4+ T cells, which produce IL-17A as their signature cytokine (Th17 cells), was detected in the draining lymph nodes of wildtype (WT) mice with EAE, as well as in the PBMCs, cerebrospinal fluid, and CNS infiltrates of patients with MS [59,60]. Deficiency of IL-23, a monokine that drives the expansion of Th17 cells and stabilizes their phenotype, conferred absolute resistance against EAE in actively immunized animals [61]. Consequently, the dogma was revised to portray Th17, as opposed to Th1 cells, as the key encephalitogenic effectors [62,63].

An in-depth examination of the EAE literature indicates that the dogma should be revised once again. In fact, the contention that a single, stereotyped Th subset is exclusively responsible for perpetuating autoimmune demyelinating disease is an over simplification. Hence, fully blown EAE is restored in IL-23 deficient mice only once they are injected intraperitoneally with recombinant IL-12 from days 0 to 8 post-immunization and intracerebrally with an IL-23 expressing viral vector 2 days prior to clinical onset [61]. Similar to C57BL/6 IFNγ knockout mice, C57BL/6 IL-17A knock-outs readily succumb to EAE following active immunization [64]. IL-17F is closely related to IL-17A, raising the possibility that it plays a compensatory role in IL-17A knockouts. However, myelin peptide /CFA-immunized, C57BL/6 IL-17F-deficient mice are also fully susceptible to EAE, even when treated with antagonistic anti-IL-17A monoclonal antibodies [64]. To further complicate matters, the Th17 cells in myelin-primed C57BL/6 wildtype (WT) mice are plastic. Fate mapping studies have demonstrated that the majority of those Th17 cells spontaneously convert to a Th1-like phenotype (via the downregulation of RORγt and IL-17 and upregulation of T-bet and IFNγ) by the time they accumulate the CNS [65]. The transformation of the Th17 cells into so-called “exTh17 cells” is dependent on exposure to IL-23 in recipient mice.

### 3.2. Independent Forms of EAE Induced by CD4+ Th1 and Th17 Cells

The vast majority of the knockout studies cited above employed the same exact model of EAE, in which inbred C57BL/6 mice are immunized with a peptide fragment of myelin oligodendrocyte glycoprotein (MOG_35–55_) emulsified in Freund’s Complete Adjuvant (CFA) and injected systemically with a heat-killed *Bordetella pertussis* toxin. Strain-specific genetic factors, as well as the artificiality of the active immunization protocol (including the use of the *Bordetella pertussis* toxin, the mechanism of which has yet to be fully elucidated), make it imprudent to extrapolate the findings made using this singular model into universal principles regarding CNS autoimmunity.

Indeed, adoptive transfer studies have substantiated that bona fide, stable Th1 and Th17 cells are independently capable of inducing EAE, and that they do so by evoking distinct cellular and molecular pathways. This was originally demonstrated using lymph node cells harvested from SJL/J mice immunized with an immunogenic peptide of proteolipid protein (PLP_139–155_) emulsified in Freund’s Incomplete (as opposed to Complete) Adjuvant [49]. Inactivated *Mycobacterium tuberculosis* was deliberately omitted from the emulsion in order to avoid bias towards a particular Th lineage during priming. Ten days following immunization, the lymph node cells were cultured *in vitro* with antigens in the presence of either recombinant IL-12 (to generate Th1 cells), IL-23 (to generate Th17 cells), or an anti-IL-12/IL23 neutralizing antibody (to generate uncommitted Th cells) prior to adoptive transfer into naïve syngeneic recipients. IL-12- and IL-23-polarized PLP_139–155_ reactive CD4+ T cells induced clinically indistinguishable forms of EAE with respect to day of onset and the kinetics and severity of the disability progression. The unpolarized cells were innocuous. Unlike their C57BL/6 MOG_35–55_ reactive counterparts, SJL PLP_139–155_-reactive Th17 donor cells are not plastic. They continue to produce IL-17, and not IFNγ, after infiltrating the CNS of naïve syngeneic hosts.

Despite their clinical similarities, the Th1- and Th17-driven SJL/PLP_139–151_ models differ with respect to the spatial distribution and cellular composition of neuroinflammatory infiltrates and the pro-inflammatory milieu in the inflamed spinal cords. Th1-mediated EAE was characterized by submeningeal MHC Class II^hi^ macrophage-rich infiltrates and CNS upregulation of CXCL9, CXCL10, CXCL11, and NOS2 mRNA, while Th17-mediated EAE was characterized by neutrophil-rich infiltrates that extended into the white matter parenchyma in finger-like projections and CNS upregulation of CXCL1, CXCL2, and granulocyte colony-stimulating factor (G-CSF). Importantly, the two forms of EAE exhibited distinct response profiles to a panel of immunomodulatory therapies. Treatment with neutralizing antibodies to IL-17 or granulocyte/macrophage colony-stimulating factor (GM-CSF) inhibited EAE induced by the transfer of IL-23-polarized, but not IL-12-polarized, cells. In contrast, neutralizing antibodies to tumor necrosis factor (TNF)α suppressed, while neutralizing antibodies to IFNγ exacerbated, both forms of EAE. Axtell et al. subsequently reported that IFNβ therapy ameliorates EAE mediated by C57BL/6, MOG_35–55_-reactive Th1 cells, but exacerbates disease mediated by their Th17 counterparts [66]. Collectively, these studies provide proof of the principle that Th1 and Th17 cells can mediate clinically indistinguishable forms of inflammatory demyelinating disease in immunocompetent hosts, which differ in histopathological features and responsiveness to individual disease-modifying agents. By extension, the variable responsiveness of patients with MS to individual disease-modifying agents that has been observed in the clinical setting might reflect inter-subject differences in underlying immunopathogenic mechanisms.

The capacity of highly polarized Th1 and Th17 cells to mediate EAE independent of one another has been definitively demonstrated using experimental systems completely devoid of IL-23 or IL-12, respectively. Purified CD4+ T cells harvested from MOG_35–55_-primed, C57BL/6 mice deficient in the common IL-12 p40 chain (which are unable to synthesize either bioactive IL-12 or IL-23) were challenged ex vivo with antigens in the presence of recombinant IL-12 or IL-23 prior to transfer into naïve IL-12 p40 deficient recipients. These “pure” Th1 and Th17 populations were equally effective at inducing EAE with high incidence and caused significant demyelination and axonopathy at inflamed foci [67,68]. Similarly, IL-12-polarized MOG_35–55_-specific Th1 cells derived from IL-23 receptor-deficient donors and IL-23-polarized Th17 cells derived from IL-12 receptor-deficient donors were shown to be encephalitogenic upon transfer into C57BL/6 WT hosts. IL-23-polarized T cells derived from IL-12p40- or IL-12 receptor-deficient donors expressed a high ratio of IL-17 to IFN-γ and high levels of RORγt in the CNS of IL-12p40 knockout and WT recipients, respectively, demonstrating that the transferred cells had steadily maintained a Th17 phenotype yet were still fully capable of mediating clinical EAE. Consistent with these results, MOG_35–55_-specific Th17 cells, generated from C57BL/6 T-bet-deficient donors, are capable of inducing full blown EAE despite the fact that they are phenotypically stable [69,70]. Hence, Th17 cells do not need to transition into an “ex-Th17” state in order to be encephalitogenic.

### 3.3. Heterogeneity of Encephalitogenic Murine T Cells with Respect to Th Phenotype and Dependence on Individual Effector Cytokines

The diversity of encephalitogenic T cells extends to their reliance on individual Th effector cytokines to induce neuroinflammation and/or damage to CNS tissue. As mentioned above, EAE mediated by IL-23-polarized PLP_139–155_-reactive Th17 cells in SJL hosts is abrogated by the neutralization of IL-17 and exacerbated by the neutralization of IFNγ [49]. In contrast, IL-23-modulated MOG_35–55_-reactive CD4+ T cells transfer comparable disease to C57BL/6 IL-17 receptor-deficient (IL-17R^−/−^) and WT hosts [71]. Surprisingly, disease in the IL-17R^−/−^ hosts is IFNγ-dependent, which may be explained by the plasticity of the C57BL/6 Th17 cells and their propensity to acquire Th1 characteristics following transfer.

As will be discussed in detail in the next section, IL-12-polarized, IFNγ-deficient effector T cells induce an atypical form of EAE in WT hosts. However, they induce conventional EAE (characterized by ascending paralysis and large spinal cord infiltrates) in IL-17R-deficient hosts [72]. Treatment of the IL-17RA-deficient adoptive transfer recipients with neutralizing antibodies against IFN-γ had no impact on the EAE severity or incidence, demonstrating that host-derived IFNγ is not responsible for the conventional EAE phenotype. In this unique IFNγ/IL-17 independent model of EAE, spinal cord infiltrates contain a relatively high proportion of neutrophils and elevated levels of GM-CSF, CXCL1, IL-1β, and CCL22. Clinical disease is curtailed by treatment with anti-GM-CSF neutralizing antibodies or anti-CXCR2 blocking antisera, but not with an IL-1 receptor antagonist [72]. Of note, GM-CSF also promotes chronic neurological disability in C57BL/6 WT recipients of either WT Th1 or Th17 cells [73], but is dispensable for Th1-mediated EAE in SJL mice [49].

Th2 cells have been shown to be innocuous and even to play an immunoregulatory role in conventional murine EAE models [47,48,74]. Th2 cytokines have often been viewed as “anti-inflammatory” in the context of EAE and MS. In fact, systemic treatment with recombinant IL-4 ameliorates EAE in SJL adoptive transfer recipients (ostensibly by shifting the phenotype of the donor cells from a Th1 to a Th2 phenotype) [75], and in actively immunized C57BL/6 mice (by diverting the migration of myelin-reactive Th17 cells from the CNS to the gut) [76]. Nonetheless, Th2 cells have a latent encephalitogenic potential that is unmasked under certain environmental conditions. For example, murine Th2 cells trigger an alternative form of EAE, with white matter infiltrates composed largely of polymorphonuclear and mast cells, upon transfer into immunodeficient Recombination-Activating Gene (RAG)-1 knockout hosts [48]. MOG-immunized marmosets developed a progressive, lethal form of EAE, associated with a shift from a Th1- to a Th2-like pattern of cytokine production, following the cessation of therapy with a soluble, “tolerogenic” formulation of MOG [77]. This severe demyelinating syndrome was characterized by plasma cell-rich white matter infiltrates and high titers of circulating anti-MOG antibodies, suggestive of IL-4- or IL-5-mediated pathology. 

The spectrum of Th cells with encephalitogenic potential is not confined to traditional Th lineages. Stimulation of Th2 cells with TGFβ drives their transformation into novel Th9 cells that produce IL-9 and IL-10 [78]. Myelin-reactive Th9 cells that express a transgenic T-cell receptor specific for MOG_35–55_ acquire the capacity to induce conventional EAE following two rounds of *in vitro* activation [79]. Th9-mediated EAE differs from Th1- or Th17-mediated diseases in that it causes peripheral nervous system lesions in the dorsal nerve roots, in addition to mononuclear infiltrates and extensive demyelination in the spinal cord white matter. Together, the above observations underscore the fact that the Th lineage commitment and mechanism of action of murine encephalitogenic T cells are heterogeneous. Furthermore, EAE susceptibility is determined by a complex interaction between the intrinsic characteristics of the autoreactive T cells and the permissiveness of the ambient environment.

### 3.4. Atypical Forms of EAE that Target the Posterior Fossa as Opposed to the Spinal Cord

EAE normally manifests as an ascending paralysis secondary to inflammatory demyelination targeted to the lumbosacral spinal cord. An alternative form of EAE has been observed in which afflicted mice exhibit imbalance/vestibular dysfunction in association with neutrophil-rich lesions in the brainstem and/or cerebellum [80,81,82,83,84]. This phenotype occurs most reliably and prominently under circumstances in which IFNγ signaling is suppressed. Hence, C57BL/6 IFNγ-deficient and IFNγ receptor (IFNγR)-deficient mice exhibit a higher incidence of atypical EAE than their wildtype (WT) counterparts following active immunization with MOG_35–55_ [54,85]. In adoptive transfer models, either deficient IFNγ production by encephalitogenic donor T cells, or impaired IFNγ signaling into host cells, is sufficient for the development of atypical EAE [59,84,86]. Atypical disease has also been observed in mice on the C3H background when injected with encephalitogenic T-cell lines that contain a high ratio of Th17 to Th1 cells, while C3H T-cell lines that contain a relatively high percentage of Th1 cells induce conventional EAE [83]. In both the C57BL/6 and C3H transfer models, atypical disease is dependent on IL-17 production/ signaling and CXCR2 chemokine-driven neutrophil recruitment [72,83,87]. This is not surprising, because a major function of IL-17 is to upregulate the expression of neutrophil-mobilizing/activating factors, such as G-CSF, and chemokines that target granulocytes, such as CXCL1, CXCL2, and CXCL5 [88]. Conversely, IFNγ skews myeloid cell differentiation in the bone marrow to favor monocytes over granulocytes during immune activation [89], and monocytes are more prevalent in the infiltrates of mice with conventional disease [82,86,87,90].

## 4. The Cytokine Profiles of T Cells in Individuals with MS

### 4.1. Myelin Antigen-Specific IFNγ and IL-17 Producing T Cells

In order to analyze the Th1/Th17 cytokine profiles of autoreactive T cells in MS, Carbajal and colleagues collected PBMCs from a cohort of untreated, relapsing-remitting patients with moderate neurologically disability on a monthly basis [68]. The frequencies of PBMCs that secreted IFNγ and/or IL-17 in response to challenges with whole human myelin basic protein (MBP) were quantified byEnzyme-Linked Immunospot (ELISpot) assays. Interestingly, the ratio of MBP-specific IFNγ to IL-17 producers was remarkably stable over the course of a year in approximately half of the subjects. The investigators found that 23% of the subjects consistently mounted IFNγ-skewed responses, 17% consistently mounted IL-17-skewed responses, and the remainder had comparable or oscillating frequencies of IFNγ and IL-17 producers. All of the subjects underwent regular assessments of cerebral lesion burden by magnetic resonance imaging. The T1 lesion load, which correlates with severe CNS injury and axonal loss, was highest in patients with the mixed IFNγ/IL-17 pattern, suggesting that autoreactive Th1 and Th17 cells might act synergistically when inflicting CNS damage. Elevated frequencies of MBP-specific IFNγ to IL-17 PBMC have also been detected in patients in a later, progressive stage of MS [43,44]. In a separate study, patients with spinal-cord predominant MS exhibited relatively high ratios of IL-17- to IFNγ-producing PBMCs in response to challenges with either MBP or MOG [91].

### 4.2. T Cells with Mixed Th1 and Th17 Characteristics

In a recent study, IFNγ/ GM-CSF expressing CD4+ T cells that express a mixture of Th1- and Th17-related molecules, (i.e., the chemokine receptors, CXCR3 and CCR6, and the transcription factors, RORγt and T-bet) were enriched in the cerebrospinal fluid and brain specimens of patients with MS [92]. An independent laboratory found that IFNγ/IL-17-co-expressing CD4+ T cells are abundant in post-mortem MS lesions, and that lymphocytes obtained from relapsing MS patients have a propensity to expand into IFNγ/IL-17 co-producers [93]. Transcriptional analyses have shown increased levels of mRNA-encoding CXCR3 and reduced levels of IFNγ, CCL3, CCL4, Granzyme B, and IL-10, in IFNγ/IL-17-co-producing CD4+ T cells sorted from the PBMC of untreated, relapsing MS patients versus age- and sex-matched healthy controls [94]. IFNγ/IL-17-co-producing CD4+ T cells from clinically stable patients expressed higher levels of IL-10 compared with the analogous cells from active patients who had developed a gadolinium-enhancing MRI lesion or a clinical relapse within 6 months of phlebotomy. Whether or not human IFNγ/IL-17-co-producing T cells represent a transitional stage between Th17 and putative ex-Th17 cells remains to be demonstrated.

## 5. Conclusions

Great strides have been made in the treatment of relapsing forms of MS. Over 15 disease-modifying agents are currently in clinical use that significantly reduce annualized relapse rates. Many of these drugs specifically target lymphocytes, but they do not distinguish between beneficial and pathogenic subsets, thereby increasing the risk of opportunistic infection. The next step in the evolution of MS therapeutics will be to develop drugs that preferentially deplete or inactivate encephalitogenic effectors without curtailing protective immunity. This will only be possible with an investment in future research that scrutinizes the characteristics of encephalitogenic effectors versus innocuous T cells using a combination of transcriptomic, proteomic, and metabolomics approaches. An increased understanding of the pathways by which encephalitogenic T cells migrate across the blood–brain barrier, expand in the CNS microenvironment, and inflict damage to CNS tissues will inevitably lead to the discovery of distinguishing biomarkers and therapeutic targets. However, this endeavor will be further complicated by the diversity of myelin-reactive T cells with encephalitogenic potential, indicating that therapeutic approaches will have to be customized for clinical subsets of patients.

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
