# Peer review of "The Diversity of Encephalitogenic CD4+ T Cells in Multiple Sclerosis and Its Animal Models"

_jcm, 2019, doi:10.3390/jcm8010120_

Reviewer 1 Report

This is an extremely well-written review provided by an outstanding clinician/scientist that has made important conceptual advances in the understanding of potential pathogenic mechanisms contributing to MS. The concise article delves into an important topic and will be well-received by both clinicians and researchers.

Author Response

We appreciate the reviewer's generous comments.  No revisions were requested.

Reviewer 2 Report

It's a nice review, very detailed and interesting for reading, includes all the important aspects of EAE and MS, but it could involve some more details to make the whole picture about MS and the future therapy more complete.

 Page 1 Line1-3 Title Since the authors discuss CD4(+) T cell-mediated MS and EAE, it should be considered to revise the title. Myelin antigen-specific CD8+ T cells can also be encephalitogenic.

Page 1 Line 9 in Abstract: Studies that have been characterizing the mechanisms used by CD8 T cells to induce CNS autoimmunity, which  indicates that myelin-specific CD4 T cells are likely to be one of several important mechanisms that drive CNS disease in MS patients.

Page 1 Line 27: Needed reference here.

Page 1 Line 32: Is it possible for authors to incorporate in their review clinically isolated syndromes as an independent risk factor for developing multiple sclerosis. The 2017 McDonald criteria continue to apply primarily to patients experiencing a typical clinically isolated syndrome. Please use the newest reference: Diagnosis of multiple sclerosis: 2017 revisions of the McDonald criteria.

Page 1 Line 33: These sentence looks like it defines a definitive list of symptoms. Please try to revise and describe these symptoms as most often or usual (include etc.,)..it is perhaps more appropriate to use i.e., sensory and visual disturbances, motor impairments, fatigue, pain and cognitive deficits, etc.

Page 1 Line 34: Extensive gray matter involvement has been associated with cognitive decline, motor deficits, fatigue, painful syndromes, and ocular motility disturbances in MS, not only white matter.

Page 1 Line 45: Try to emphasize the importance of B cells in immunopathogenesis although the review is based only on T cell involvement. In addition, please involve Microglia and astrocytes, which may create a survival niche for long-term plasma cell survival. Parker Harp CR, et al.

PLoS One. 2018;13(6):e0199694.

It has been accepted that MS is an autoimmune disease mediated by T cells, which is primarily based on experimental evidence from rodent models.  However, B cells in the development of MS is increasingly indicated from the study of responses to immunomodulatory therapies in MS (Baker et al., 2017, Lehmann-Horn et al., 2017, Hauser et al., 2017, Ceronie et al. 2018): Pryce G, Baker D., Mult Scler Relat Disord. 2018;25:131-137.

Can you incorporate oligoclonal bands in the context of clinically isolated syndromes too? Taken from the newest reference by Arrambide G, Brain. 2018 Apr 1;141(4):1075-1084.

Page 2 Line 58-73 The classical animal model of MS, CD4 T cell mediated EAE (CD4–EAE) predominantly induces Th1 and Th17 phenotype CD4 T cell responses, and is a poor elicitor of CD8 responses and the disease pathologies associated with CD4–EAE are usually less diverse than the disease symptoms in MS patients. CNS-specific CD8 T cells are present in the peripheral T cell repertoire and when activated, can induce CNS autoimmunity. Although, the MHC class I alleles are to a lesser extent associated with MS, it is proper to mention here in in this part “The autoimmune hypothesis”  this fact before describing only the CD4+ mediated EAE or MS, not to underestimate CD8–EAE  (MHC I) mediated CNS autoimmunity.

 Page2 Line 51: Needed reference.

Author Response

We appreciate the reviewer's insightful comments.  We have revised the manuscript based on the reviewer's critique .  A point-by-point response follows:

Page 1 Line1-3 Title Since the authors discuss CD4(+) T cell-mediated MS and EAE, it should be considered to revise the title. Myelin antigen-specific CD8+ T cells can also be encephalitogenic.

We revised the title to specify "...Encephalitogenic CD4+ T cells..."

Page 1 Line 9 in Abstract: Studies that have been characterizing the mechanisms used by CD8 T cells to induce CNS autoimmunity, which  indicates that myelin-specific CD4 T cells are likely to be one of several important mechanisms that drive CNS disease in MS patients.

We changed the wording of the abstract to indicate that CD4+ T cells are one of several immune effector subsets implicated in MS pathogenesis, and that the different leukocyte subsets likely act in concert.  (Page 1, lines 10-11).

Page 1 Line 27: Needed reference here.

A reference was added (reference # 1 in the revised manuscript).

Page 1 Line 32: Is it possible for authors to incorporate in their review clinically isolated syndromes as an independent risk factor for developing multiple sclerosis. The 2017 McDonald criteria continue to apply primarily to patients experiencing a typical clinically isolated syndrome. Please use the newest reference: Diagnosis of multiple sclerosis: 2017 revisions of the McDonald criteria.

We discuss the 2017 McDonald criteria in greater detail, and specifically reference the more recent seminal article that  introduced the revised criteria (Page 1, lines 31-39 and Page 2, lines 54-56; reference #6).

Page 1 Line 33: These sentence looks like it defines a definitive list of symptoms. Please try to revise and describe these symptoms as most often or usual (include etc.,)..it is perhaps more appropriate to use i.e., sensory and visual disturbances, motor impairments, fatigue, pain and cognitive deficits, etc.

In the revised manuscript, we explicitly describe the most common symptoms of MS, and make it clear that the clinical manifestation of MS is heterogeneous (Page 1, lines 39-42). 

Page 1 Line 34: Extensive gray matter involvement has been associated with cognitive decline, motor deficits, fatigue, painful syndromes, and ocular motility disturbances in MS, not only white matter.

We now discuss gray matter involvement in MS, and its relationship to cognitive impairment (Page 1, lines 43-45; reference # 7).

Page 1 Line 45: Try to emphasize the importance of B cells in immunopathogenesis although the review is based only on T cell involvement. In addition, please involve Microglia and astrocytes, which may create a survival niche for long-term plasma cell survival. Parker Harp CR, et al.

        PLoS One. 2018;13(6):e0199694.

        It has been accepted that MS is an autoimmune disease mediated by T cells, which is                     primarily based on experimental evidence from rodent models.  However, B cells in the                     development of MS is increasingly indicated from the study of responses to                                        immunomodulatory therapies in MS (Baker et al., 2017, Lehmann-Horn et al., 2017, Hauser et         al., 2017, Ceronie et al. 2018): Pryce G, Baker D., Mult Scler Relat Disord. 2018;25:131-137.

        Can you incorporate oligoclonal bands in the context of clinically isolated syndromes too?         Taken from the newest reference by Arrambide G, Brain. 2018 Apr 1;141(4):1075-1084.

A discussion on the putative role of B cells in MS, the efficacy of B cell depleting therapies in suppressing MS relapses, and relevant references were added to the manuscript (Page 2, lines 54-67; reference #'s 8-17)

Page 2 Line 58-73 The classical animal model of MS, CD4 T cell mediated EAE (CD4–EAE) predominantly induces Th1 and Th17 phenotype CD4 T cell responses, and is a poor elicitor of CD8 responses and the disease pathologies associated with CD4–EAE are usually less diverse than the disease symptoms in MS patients. CNS-specific CD8 T cells are present in the peripheral T cell repertoire and when activated, can induce CNS autoimmunity. Although, the MHC class I alleles are to a lesser extent associated with MS, it is proper to mention here in in this part “The autoimmune hypothesis”  this fact before describing only the CD4+ mediated EAE or MS, not to underestimate CD8–EAE  (MHC I) mediated CNS autoimmunity.

 In the revised manuscript, we acknowledge the fact that CD8+ T cells can induce EAE and that CD8+ T cells are present in MS lesions. (Page 2, lines 85-89; references # 21 and 22).

Page2 Line 51: Needed reference.

A reference has been added, as suggested (reference # 19).

This manuscript is a resubmission of an earlier submission. The following is a list of the peer review reports and author responses from that submission.